# MDM2-Dependent Rewiring of Metabolomic and Lipidomic Profiles in Dedifferentiated Liposarcoma Models

**DOI:** 10.3390/cancers12082157

**Published:** 2020-08-04

**Authors:** Andrew Patt, Bryce Demoret, Colin Stets, Kate-Lynn Bill, Philip Smith, Anitha Vijay, Andrew Patterson, John Hays, Mindy Hoang, James L. Chen, Ewy A. Mathé

**Affiliations:** 1Department of Biomedical Informatics, The Ohio State University, Columbus, OH 43210, USA; patt.14@osu.edu; 2Division of Preclinical Innovation, National Center for Advancing Translational Sciences, NIH, 9800 Medical Center Dr., Rockville, MD 20892, USA; 3Biomedical Sciences Graduate Program, The Ohio State University, Columbus, OH 43210, USA; 4Department of Internal Medicine, The Ohio State University, Columbus, OH 43210, USA; bd550418@ohio.edu (B.D.); colin.stets@osumc.edu (C.S.); katelynn.bill@gmail.com (K.-L.B.); John.Hays@osumc.edu (J.H.); mindy.hoang@osumc.edu (M.H.); 5The Huck Institutes of Life Sciences, Penn State University, State College, PA 16802, USA; pbs13@psu.edu (P.S.); azv2@psu.edu (A.V.); adp117@psu.edu (A.P.)

**Keywords:** dedifferentiated liposarcoma (DDLPS), metabolomics, lipidomics, MDM2, sphingolipid metabolism, atorvastatin, chemosensitivity

## Abstract

Dedifferentiated liposarcoma (DDLPS) is an aggressive mesenchymal cancer marked by amplification of MDM2, an inhibitor of the tumor suppressor TP53. DDLPS patients with higher MDM2 amplification have lower chemotherapy sensitivity and worse outcome than patients with lower MDM2 amplification. We hypothesized that MDM2 amplification levels may be associated with changes in DDLPS metabolism. Six patient-derived DDLPS cell line models were subject to comprehensive metabolomic (Metabolon) and lipidomic (SCIEX 5600 TripleTOF-MS) profiling to assess associations with MDM2 amplification and their responses to metabolic perturbations. Comparing metabolomic profiles between MDM2 higher and lower amplification cells yielded a total of 17 differentially abundant metabolites across both panels (FDR < 0.05, log2 fold change < 0.75), including ceramides, glycosylated ceramides, and sphingomyelins. Disruption of lipid metabolism through statin administration resulted in a chemo-sensitive phenotype in MDM2 lower cell lines only, suggesting that lipid metabolism may be a large contributor to the more aggressive nature of MDM2 higher DDLPS tumors. This study is the first to provide comprehensive metabolomic and lipidomic characterization of DDLPS cell lines and provides evidence for MDM2-dependent differential molecular mechanisms that are critical factors in chemoresistance and could thus affect patient outcome.

## 1. Introduction

Dedifferentiated liposarcoma (DDLPS) is a highly morbid, adipocytic tumor accounting for approximately 20% of all soft-tissue sarcomas [1]. Liposarcomas are generally thought to arise spontaneously rather than from preexisting benign lesions, and most patients lack recognized causative factors. Although complete surgical resection can be curative, DDLPS often develops in deep anatomic locations, such as the retroperitoneum or mediastinum, where its propensity to encase vital structures typically renders a complete surgical resection difficult or impossible. In fact, the five year survival of patients with these abdominal liposarcomas is only 20% [2]. Unfortunately, chemotherapy has limited efficacy in the treatment of DDLPS, with single-agent response rates of up to 30% [3]. Systemic therapeutic regimens improve survival only modestly when complete surgical resection is not feasible [3]. Improved treatments are critically needed for this highly morbid disease.

At the molecular level, DDLPS is characterized by an amplification of the 12q portion of the chromosome resulting in excess copies of the mouse double minute 2 homolog (MDM2) [4,5,6,7]. MDM2 inhibits the tumor suppressor TP53. Thus, an amplification in MDM2 results in a shift towards pro-growth pathways. Our previous work demonstrated that higher levels of MDM2 amplification are associated with worsened overall survival and resistance to DNA-damaging chemotherapy in liposarcomas [8,9]. Interestingly, although DDLPS are of fat origin, they produce scant fat themselves and resemble undifferentiated pleomorphic or spindle cell sarcoma, typically showing moderate or high cellularity, with moderate to marked pleomorphism [10]. Given these observations, we hypothesized that central to the oncogenicity of DDLPS is its ability to alter fat metabolism and other key metabolic pathways in an MDM2-dependent manner. We thus performed a comprehensive metabolomic and lipidomic profiling of patient-derived DDLPS cell lines with varying degrees of MDM2 amplification. Taken together, this is the first attempt to characterize the metabolome and lipidome of DDLPS in light of each cell line’s genomic amplification milieu.

## 2. Results

To test our hypothesis, that treatment impact on DDLPS tumors differs for tumors with higher vs. lower MDM2 amplification, we measured the metabolomic and lipidomic effects of MDM2 amplification, MDM2 inhibition, cholesterol inhibition, and ceramide treatment in six patient-derived DDLPS cell lines (Table 1). DDLPS cell lines were categorized as MDM2 lower and higher amplification based on prior published PCR analyses [11]. Lipo-224, LPS141, and Lipo-246 cells have higher MDM2 amplification, and Lipo-815, Lipo-224B, and Lipo-863 have lower MDM2 amplification (Table 1, Appendix A). MDM2 amplification levels were verified by DNA copy number and RNA transcription levels, showing high concordance (Pearson’s r = 0.92, P = 0.03) [11]. Treatment of the cell lines with varying doses of doxorubicin confirmed that MDM2 higher cell lines had lower chemosensitivity than MDM2 lower cells (Figure 1a).

### 2.1. Metabolomic Changes Associated with MDM2 Amplification Levels

Metabolomic profiles were generated using the Metabolon platform in at least triplicate measurements. Appendix A provides a summary of the conditions used. A total of 541 metabolites were measured (including 62 Metabolomics Standard Initiative (MSI) Level 1 identified metabolites [13]). Measured metabolites include lipids, amino acids, nucleotides, carbohydrates, peptides, cofactors and vitamins, xenobiotics, and energy metabolites, as defined by the Metabolon “superpathway” designation (Figure 1b). Unsupervised clustering of samples by their metabolomic profiles (Appendix A) showed that biological replicate samples cluster closely together, providing confidence in the reproducibility of the measurements.

When comparing metabolite levels between MDM2 higher and lower cells, we identified 17 altered metabolites (FDR-adjusted *p* value < 0.05, log2 fold change (FC) > 0.75), of which 15 were elevated in MDM2 higher cells and two were elevated in MDM2 lower cells (Appendix A). Of these, we noted that lipids constitute 41% (seven species) of the metabolites altered, and the rest comprise three amino acids, three peptides, two nucleotides, one vitamin- and cofactor-associated metabolite, and one energy-associated metabolite (Figure 1c and Appendix A). Interestingly, pathway analysis revealed that sphingolipid metabolism/de novo biosynthesis were the top enriched pathways. Additional pathways prioritized included synthesis of prostaglandins and thromboxanes, beta-oxidation, and bile acid synthesis/secretion (Appendix A and Figure 1d).

### 2.2. Raising the MDM2 Levels in DDLPS Lower Cell Lines Results in Induction of Key Metabolites Resembling Those of MDM2 Higher Cells

We previously demonstrated that MDM2-TP53 binding inhibitors temporarily result in further amplification of MDM2 by allowing TP53 to induce MDM2 expression [11]. Thus, MDM2-TP53 binding inhibitors temporarily increase MDM2 levels in both MDM2 higher and lower amplification DDLPS cell lines (Appendix A). We evaluated the metabolomic profiles of DDLPS cell lines (LPS141, Lipo-246, Lipo-863, and Lipo-815) with and without MDM2-TP53 binding inhibitor RG7112. When comparing the effect of MDM2 inhibitor treatment in all cell lines (treated vs. untreated) regardless of MDM2 status, no significant metabolites were identified (Appendix A).

When we considered the effect of MDM2 inhibitor treatment in MDM2 higher and lower cells separately, we observed six altered metabolites when comparing MDM2 lower cells before and after treatment. Of these six, four were upregulated by treatment and two were downregulated (Figure 2a,b and Appendix A). No metabolites were altered in MDM2 higher cells (Figure 2c). The six metabolites altered by MDM2 binding inhibition in MDM2 lower cells comprised three lipids (1-(1-enyl-palmitoyl)-2-oleoyl-Glycerophosphoethanolamine (P-16:0/18:1), myristate (14:0), palmitoylcholine) and three nucleotides (adenylosuccinate, thymidine 5${}^{\prime }$-monophosphate, uridine). Pathways associated with these metabolites in the RaMPpathway database [12] included pyrimidine salvage, pyrimidine catabolism, and nucleotide salvage.

### 2.3. Independent Lipidomic Profiling Confirms Lipids Found to Be Altered by MDM2 Amplification from Metabolomic Analysis and Identifies Additional Relevant MDM2-Dependent Lipids

Given the larger numbers of lipids altered by MDM2 with our metabolomic analyses, we performed an independent lipidomic analysis (SCIEX 5600 TripleTOF-MS) in the same DDLPS cell line models grown analogously to the prior metabolomic profiling experiments. The quality of the experiment was assessed through PCA analysis and showed that pooled QC samples, blanks, and biological replicates clustered tightly together (Appendix A). The lipid coverage in our metabolomic and lipidomic profiling was carefully assessed to identify common lipids covered by both platforms. Of all lipids measured, twenty-three percent were only identified in the Metabolon platform, 67% only in the lipidomic analysis, and 10% in both (Appendix A). The correlation between the alterations of the ratios of abundance in MDM2 higher vs. lower cells in identical lipid species from both platforms was high (Pearson ρ = 0.67, Appendix A), demonstrating that the measurements are reproducible and robust.

A total of 433 lipids were measured and identified, and lipid classes included a wide variety of glycerophospholipids such as glycerophosphocholines, glycerophosphoethanolamines, and glycerophosphoinositols, glycerolipids such as diradylglycerols and triacylglycerols, as well as sphingolipids such as sphingomyelins and ceramides (Figure 3a). When comparing lipid levels between MDM2 higher and lower amplification cells, seven lipids were altered, six of which were elevated in MDM2 low cells (monogalactosyldiacylglycerol (16:0_22:6), plasmenyl-Phosphatidylcholine (PC) (P-18:0/22:5), plasmenyl-PC(P-18:1/20:4), Sphingomyelin (SM) (d18:0/24:0), SM(d18:1/24:1), SM(d22:1/22:1)) and one of which was elevated in MDM2 high cells.

The results of the MDM2 higher vs. lower comparison in the lipidomic panel provided further evidence for perturbation of the sphingolipid metabolism pathway. Three sphingomyelins were elevated in MDM2 lower cells compared to MDM2 higher cells, in addition to two plasmalogens and one glycosyldiradylglycerol. Additionally, one hexosylceramide non-hydroxyfatty acid-sphingosine (HexCer_NS), (HexCer_NS (d18:1/16:0), *p* = 0.02) was significantly upregulated in MDM2 higher cells. This mirrored the results of the metabolomic analysis, where the same glycosylated ceramide (glycosyl-N-palmitoyl-sphingosine, *p* = 0.03) was also upregulated in MDM2 high compared to low. Three other glycosylated ceramides were borderline significant, showing log2 fold changes > 0.75, but unadjusted *P*-values < 0.05. When specifically assessing glycosylated ceramides and ceramides, we show that elevation of glycosylated ceramides in MDM2 higher cells is confirmed in both platforms (Appendix A).

### 2.4. Induction of the Sphingolipid Pathway in DDLPS Models Using Atorvastatin Resulted in Chemoresistance

In previous literature, 3-hydroxy-3-methyl-glutaryl-CoA reductase reductase inhibitor (statin) treatment was noted to result in ceramide elevation [14,15,16]. Given the importance of the ceramide metabolites in our findings, we wanted to explore whether induction of ceramides altered chemosensitivity. To this end, we used the commonly prescribed HMG-CoA reductase inhibitor, atorvastatin. We observed that the growth of MDM2 higher cells was not affected by atorvastatin treatment, while MDM2 lower cells exhibited slower growth (Figure 3b). All DDLPS cell lines showed a decrease in cell viability, although the effect was more pronounced in MDM2 lower amplification cells (Appendix A).

We then tested the effects of HMG-CoA reductase inhibition on the lipidome of both MDM2 higher and MDM2 lower cell lines (Figure 3c,d and Appendix A). As expected, very few lipids were altered in MDM2 higher cells following treatment. Conversely, we observed dramatic shifts in the lipidome of MDM2 lower cells following atorvastatin treatment, with 52 lipids upregulated and 11 downregulated in response to treatment. Altered lipids included 11 ceramides, three glycosylated ceramides, and two sphingomyelins; all increased following atorvastatin treatment. Previous studies have demonstrated that statins can exert off-target effects on the sphingolipid metabolism pathway [17], which could explain the shifts observed in our data. Given that we noted elevated glycosylated ceramides, we hypothesized that this atorvastatin-driven shift could alter the chemosensitivity of the DDLPS models.

As a proof of concept, we examined the effects of the addition of atorvastatin on the cytotoxic effects of the anthracycline doxorubicin. Specifically, we tested a series of different concentrations of both atorvastatin and doxorubicin and then measured their effect on cell viability in the Lipo-246 DDLPS cell line model (MDM2 higher amplification). The cell viability in the combinations was more potent together than what we would have expected. Using the Chou–Talalay method [18], we note that the combination indices were nearly all above one indicating antagonistic agents. While both atorvastatin and doxorubicin both had inhibitory effects on the cell line viability as individual agents, when combined, we observed a cooperativity index > 1 for nearly all combinations, consistent with antagonism (Appendix A).

### 2.5. Glycosylated Ceramides Are Consistently Elevated in MDM2 Higher Cells

Noting the elevation of glycosylated ceramides across panels in MDM2 higher cells compared to lower, we evaluated the sensitivity of the cell lines to treatment with non-glycosylated ceramides and found that the cell lines’ viability decreased, regardless of MDM2 status (Figure 4a). Glycosyl-N-palmitoyl-sphingosine, a glycosylated ceramide, showed elevated levels in MDM2 high cells. This was the same glycosylated ceramide that was also highly upregulated in MDM2 high cells in the lipidomic panel (HexCer_NS (d18:1/16:0), log2 FC = 2.36, FDR = 0.02, Figure 4b). Further, Lipid Ontology enrichment (LION/Web) [19] analysis of lipids perturbed between untreated MDM2 higher and MDM2 lower cells returned “ceramide phosphocholines (sphingomyelins)” as a significantly enriched term (Figure 4c). Lastly, to further validate differences in glycosylated ceramide levels between MDM2 higher and lower cells, we performed unsupervised clustering of cell lines in the lipidomic panel by abundance of all glycosylated ceramides (Figure 4d) and found that cells clustered completely by MDM2 amplification status.

## 3. Discussion

Prior studies have identified upregulation of nucleotide and serine synthesis [20] in liposarcoma compared to healthy cells, as well as the activity of the nucleoside salvage pathway [21]. We observed two nucleotides, three peptides, and two amino acids that were altered between MDM2 higher and lower cells, which could indicate that MDM2 higher and lower tumors could be differentially sensitive to the previously reported therapeutic targets of amino acid and nucleotide salvage and synthesis. We further observed significant differences in lipid metabolism between MDM2 higher and lower cells and MDM2-dependent shifts in cellular growth of DDLPS cells in response to modulation of lipid metabolism via statin treatment. Sphingolipid metabolism was consistently observed and is a known important therapeutic target in a variety of cancer types. For example, sphingomyelin synthase is frequently inhibited to prevent the conversion of ceramide to sphingomyelin to maintain its pro-apoptotic activity [22]. In the metabolomic panel, two lipids altered between MDM2 higher and lower cells mapped to the sphingolipid metabolism/glycosphingolipid metabolism pathways. We identified three sphingomyelins that were depleted in MDM2 higher cells through our lipidomic assay. Interestingly, while glycosylated ceramides were strong predictors of MDM2 amplification status, ceramides were not (results not shown), meaning that the ratio of glycosylated/nonglycosylated ceramides was shifted between MDM2 higher and lower cells. Glycosylated ceramides are known to be able to drive drug resistance without a change in the levels of their associated nonglycosylated form [23], counteracting the pro-apoptotic signaling exerted by ceramides [24]. This suggests a possible mechanism of resistance to atorvastatin in MDM2 higher cells driven by elevated ratios of glycosylated:nonglycosylated ceramides. These results are concordant with previous findings showing the elevation of ceramides in response to stress (e.g., chemotherapy) [25]. Sphingolipid metabolism is an important therapeutic target in a variety of cancer types. For example, sphingomyelin synthase is frequently inhibited to prevent the conversion of ceramide to sphingomyelin to maintain its pro-apoptotic activity [22]. Both our lipid and metabolite data provided evidence for the perturbation of sphingolipid metabolism in MDM2 higher cells compared to lower. These results are concordant with previous findings showing the elevation of ceramides in response to stress (e.g., chemotherapy), thereby leading to cell death [25].

We also note that MDM2 higher cells showed an increase in metabolites involved in de novo fatty acid synthesis. Many cancer types, such as breast, colorectal, and ovarian cancer, show an upregulation of fatty acid synthesis [26,27,28]. Upregulation of lipogenic enzymes that produce fatty acids has been associated with poor prognosis and resistance to chemotherapy [29]. Further, three of the lipids upregulated in MDM2 higher compared to lower cells were saturated fatty acids: myristate (14:0), stearate (18:0), and nonadecanoate (19:0). Saturated fatty acids are known drivers of chronic inflammation through TLR4/nfκB-dependent signaling [30], which is a hallmark of cancers with increased risk [31]. This hypothesis is further supported by the identification of beta-oxidation and alpha-linoleic metabolism as perturbed pathways between MDM2 higher and lower cells, which is evidence of increased lipid metabolism/synthesis. Upregulation of beta oxidation is increasingly being recognized as a consistent feature of the cancer metabolic landscape [32,33]. Previous studies have indicated that DDLPS tumors have a distinct fatty acid composition compared to other liposarcoma cell types and benign lipomas [34].

In light of the different atorvastatin treatment responses displayed by MDM2 high and low cells (Figure 3a), the shift in bile acid pathways observed between MDM2 high and low cells was a significant finding. Bile acids are steroids generated from cholesterol in the liver, undergoing further metabolism in the gut by enzymes derived from intestinal bacteria [35]. In addition to the key role they play in intestinal uptake of lipids and vitamins, bile acids act as signaling molecules that regulate cell growth, as well as inflammation [36]. As such, perturbation of bile acid synthesis is known to be a factor in a variety of cancer types, such as colorectal cancer [37], hepatocellular carcinoma [38], and cholangiocarcinoma [39].

## 4. Materials and Methods

### 4.1. In Vitro Models

The culture of human LPS cell lines (Lipo-246, Lipo-863, Lipo-815, Lipo-224, Lipo-224B) has been previously reported [40]. Dr. Jonathan Fletcher (Boston, MA) generously provided us with the LPS141 cell line. All cells were cultured in Dulbecco’s Modified Eagle’s Medium (DMEM) and supplemented with 10% fetal bovine serum (FBS), 100 U/mL penicillin, and 100 U/mL streptomycin. These cells were cultured in a humidified chamber delivering 5% CO2 at 37 °C.

### 4.2. Chemical Reagents

Doxorubicin was purchased from Cayman Biochemicals. The MDM2 inhibitor SAR405838 was purchased from Selleckchem. Atorvastatin was purchased from Sigma-Aldrich. All drugs were prepared per the manufacturers’ instructions. Serial dilutions were made to obtain final concentrations for cellular assays of DMSO not exceeding 0.01%.

### 4.3. Cell Proliferation via the MTT Assay and Cooperativity Evaluation

Exponentially growing DDLPS cell lines were seeded into 96 well plates and treated with the indicated compounds. After 24 h, treatment was added to the cells in the plate. The treatment conditions were as follows: doxorubicin only, atorvastatin only, constant atorvastatin doses paired with increasing doxorubicin doses, and constant doxorubicin doses paired with increasing atorvastatin doses. Doxorubicin was added in the following doses: 0.1 μM, 0.3 μM, 0.7 μM, 1.2 μM, 2 μM. Atorvastatin was added in the following doses: 5 μM, 10 μM, 15 μM, 20 μM, 30 μM. After 72 h, the XTT cell proliferation kit from Roche Applied Science (11465015001) was used to assess cell viability following the manufacturer’s instructions. The absorbance was measured at 470 nm. The calculation of effect size was done via Calcusyn software. Live cell imaging was performed using the Incucyte Zoom system.

### 4.4. Western Blotting

Western blots were performed using Odyssey CLx (Li-Cor) and ECL (PerkinElmer). The antibodies were used as indicated per experiment: p53, p21, β-actin (Santa Cruz); MDM2 (Abcam); cleaved caspase-3 (Cell Signaling).

### 4.5. Metabolomic and Lipidomic Data Acquisition

DDLPS were plated in 60 mm dishes, drug treated with atorvastatin or SAR405838, and cultured for 72 h. Because these treatments are not cytotoxic agents, peak changes are seen later as metabolic and genomic shifts take more time to manifest, requiring a longer time point for treatment. After treatment, cells were collected using 1 mL of cold (−20 °C) methanol and a cell scraper and then immediately stored at −80 °C. Cell suspensions were sent to Metabolon Inc. [41] for comprehensive metabolomic profiling and to the Penn State University Metabolomics Facility for lipidomic profiling.

For lipidomic profiling, cells were pelleted and processed using a chloroform:methanol homogenization followed by an isopropanol:acetonitrile extraction as previously described [42]. Samples were separated by reverse phase HPLC using a Prominence 20 UFLCXR system (Shimadzu, Columbia MD) with a Waters (Milford, MA) CSH C18 column (100 mm × 2.1 mm 1.7 μm particle size) maintained at 55 °C and a 20 m aqueous/acetonitrile/isopropanol gradient, at a flow rate of 225 μL/min. For electrospray ionization positive mode, Solvent A was 40% water, 60% acetonitrile with 10 mM ammonium formate and 0.1% formic acid, and Solvent B was 90% isopropanol, 10% acetonitrile with 10 mM ammonium formate and 0.1% formic acid. For electrospray ionization negative mode, Solvent A was 40% water, 60% isopropanol with 10 mM ammonium acetate, and Solvent B was 90% isopropanol, 10% acetonitrile with 10 mM ammonium acetate. The initial conditions were 60% A and 40% B, increasing to 43% B at 2 min, 50% B at 2.1 min, 54% B at 12 min, 70% B at 12.1 min, and 99% B at 18 min, held at 99% B until 20.0 min, before returning to the initial conditions. The eluate was delivered into a 5600 (QTOF) TripleTOF using a Duospray™ ion source (all AB Sciex, Framingham, MA, USA). The capillary voltage was set at 5.5 kV in positive ion mode and 4.5 kV in negative ion mode, with a declustering potential of 80 V. The mass spectrometer was operated in IDA (information-dependent acquisition) mode with a 100 ms survey scan from 100 to 1200 m/z and up to 20 MS/MS product ion scans (100 ms) per duty cycle using a collision energy of 50 V with a 20 V spread.

### 4.6. Metabolomic Data Preprocessing

Metabolite levels were pre-processed by Metabolon, including sample normalization by Bradford protein concentration, median scaling, and missing value imputation to minimum values. There were a total of 541 identified metabolites. Metabolites with 50% missing values or greater in all samples were removed, eliminating 33 metabolites. Metabolites with a high coefficient of variation (greater than 150) were also removed, as extremely variable metabolites may be a result of technical error rather than actual biological variation. After applying these filters, four-hundred eighty metabolites remained for further analysis.

### 4.7. Lipidomic Data Preprocessing

The Proteowizard software suite [43] was used to convert the wiff/wiff.scan raw data files to mzML/ms2 file formats, using the following (1) peakpicking filters: pickertype=cwt, signal-to-noise ratio = 0.1, and peakSpace = 0.1 and 2) conversion filters: mslevel 1-1 was used for .mzl conversion, and mslevel 2-2 was used for .ms2 conversion. Converted files were then input into MZmine2 Version 2.42 [44] for peak calling and chromatogram alignment. Mass detection with a centroid mass detector on MS Level 1 was performed. Next, the ADAP chromatogram builder was used, with a minimum scan span of 3 above a group intensity threshold of 1000, a minimum highest intensity of 1000, and an m/z tolerance of 0.005 m/z or 10 ppm. Chromatogram deconvolution was performed with the ADAP wavelet algorithm, with a minimum feature height of 1000, a coefficient/area threshold of 10, a peak duration range of 0–2, and a retention time wavelet range of 0–0.1. Next, isotopic peak grouping was performed, with an m/z tolerance of 0.005 m/z or 5 ppm, a retention time tolerance of 1% relative, and a maximum charge of 3. Next, a duplicate feature filter was applied, with a filter mode of “NEW AVERAGE”, an m/z tolerance of 0.005 m/z or 5 ppm, and a retention time tolerance of 1% relative. Join alignment was performed with an m/z tolerance of 0.009 m/z or 10 ppm, a weight for m/z of 85, a retention time tolerance of 1% relative, and a weight for RT of 85. Finally, the same RT and m/z range gap filler was applied, with an m/z tolerance of 0.005 m/z or 5 ppm. This generated csv feature tables for the positive and negative ion modes, which were used as input for peak identification using the LipidMatch software [45].

For LipidMatch lipid identification, the default Sciex parameters were used. A retention time window of 0.14 m was used. An m/z search tolerance of 0.005 Da was used for MS1, and an M/Z search window of 10 ppm was used for MS2. Isolation window was set to 1 Da. Minimum scans for fragments were set to 1. The intensity threshold for MS2 was set to 1000. The All-ion-fragmentation (AIF)minimum number of scans was set to 5. The AIF correlation cutoff (adjusted R2 correlation between precursors and fragments) was set to 0.6. The features tables were configured so that they matched the default order of columns and rows expected by LipidMatch (Column 1 is row ID; Column 2 is M/Z; Column 3 is retention time; numeric data starts on Row 2). Abundance values were normalized for each ionization mode independently using the total ion current method [46]. Normalized abundances were log2 transformed, and data from both ionization modes were merged. Further analysis was restricted to lipids identified from MS2 data. The highest abundance feature was selected for further analysis when multiple features had the same identification. If a lipid was identified in both modes, negative adducts were preferred over positive adducts. Sodium adducts were only used if matching identifications with other adducts were not found. A total of 430 unique lipids were identified and quantified.

### 4.8. Statistical Analysis

Differentially abundant metabolites and lipids between groups (e.g., MDM2 status, treatment status, etc.) were assessed using linear mixed effect modeling with cell line of origin incorporated as a random effect. Resulting P-values were corrected for multiple comparisons using the method by Benjamini-Hochberg to correct the false discovery rate (FDR) [47]. Metabolites were considered statistically significant when their associated FDR-adjusted P-values were greater than 0.05 and absolute log2 fold change > 0.75. Results of all statistical testing can be found in Appendix A.

### 4.9. Pathway Analysis

We developed a novel network-based pathway enrichment strategy that integrates pathway annotations from the RaMP database [12], as well as chemical structure similarity. Metabolites associated with phenotype (e.g., MDM2 amplification or treatment) that mapped to KEGG pathways [48,49,50] were used as the input. Ten out of 18 metabolites altered by MDM2 amplification mapped to KEGG pathways. We obtained a list of all KEGG pathways that contained at least one of these altered metabolites. We then obtained a list of all metabolites involved in this list of pathways. We used these metabolites to build a mutual pathway participation network, where nodes were metabolites and edges encoded pathway similarity between metabolites. Pathway similarity was quantified using a Jaccard index of pathway annotations shared by metabolites connected by an edge. A chemical similarity network was constructed between this same set of metabolites, where edges represented the chemical fingerprint overlap between two metabolites using a Tanimoto score. The chemical similarity network was then binarized, with edges in the 90th percentile or higher of chemical similarity receiving an edge weight of 1 and all other edge weights assigned a 0. The two separate similarity networks were then merged into a consensus network model by summing edge weights. We then searched for metabolites proximal to our set of ten or nine altered metabolites using a random walk with restarts strategy [51]. To control for the connectivity of the network structure, we ran the algorithm with 10,000 random seed sets of ten or nine and compared the proximity score of nodes with the real seed set to the distribution of random sets. Metabolites that were scored very high compared to their randomized distribution (97th percentile or higher) were extracted to form a subnetwork with the seed nodes. Lastly, we applied the Louvain clustering algorithm to identify highly interconnected clusters (e.g., red, blue, and green clusters in Appendix A). Metabolites in clusters were tested separately for pathway enrichment using RaMP (FDR-adjusted *p*-value < 0.05).

### 4.10. Code and Data Availability

Preprocessed data along with the R code used for the analysis can be found at https://github.com/andyptt21/MDM2_reprograms_DDLPS_metabolism. Raw lipidomics data were submitted to Metabolomics Workbench under study track ST001405.

## 5. Conclusions

To our knowledge, this is the first detailed examination of the metabolome/lipidome of dedifferentiated liposarcoma cell lines. As such, our experimental approach allowed us to interrogate the metabolite and lipid differences that could underlie the heightened chemoresistance of MDM2 higher tumor cells as compared to lower. Our metabolomic/lipidomic analyses uncovered a wide variety of differences between MDM2 higher and lower cells, and in response atorvastatin treatment. Many of the pathway level shifts we observed suggest that MDM2 plays a role in a wide variety of cancer-related metabolic processes, including ceramide metabolism, amino acid synthesis, lipogenesis, and inflammation. The mirrored and complementary results we observed between the metabolomic and lipidomic datasets served as the validation of our findings. These data serve as a key basis for the further development of metabolic targets in this morbid disease, as well as potential clinical work. Metabolically-driven anti-cancer therapies based on these observations may potentially spare untargeted toxicities to the patient in this difficult to treat disease. Future research focus will include examination of mouse xenograft models of DDLPS and of patient tissue for validation of our in vitro findings of perturbation in ceramide metabolism.

## Figures and Tables

**Figure 1 cancers-12-02157-f001:**
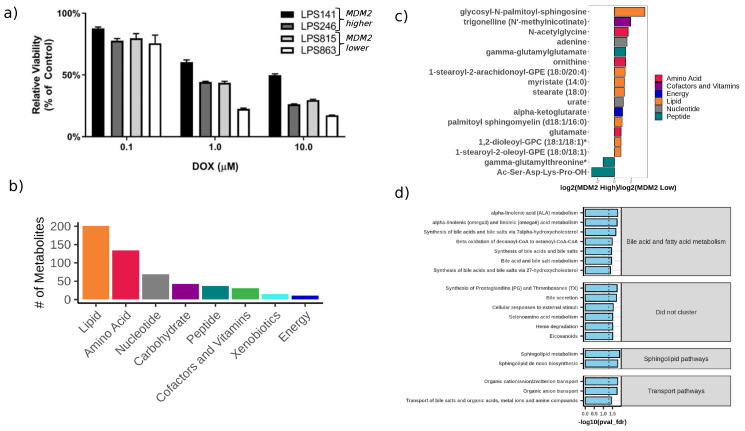
DDLPS cell lines with higher or lower MDM2 amplification show distinct chemosensitivity and metabolomic profiling. (**a**) Cell viability of DDLPS cell lines is elevated in MDM2 higher cells after doxorubicin treatment, (**b**) Distribution of metabolite classes (Metabolon superpathway) represented in the Metabolon panel. (**c**) Metabolites that are altered between MDM2 higher and lower amplification cell lines (FDR-adjusted *p*-value < 0.05 and |log2 fold change| > 0.75). (**d**) Over-represented pathways (FDR-adjusted *p*-value < 0.05) in metabolites that are altered between cells with higher vs. lower MDM2 amplification in the blue module of the final network enrichment model (Appendix A). Pathways are grouped by cluster as determined by the relational database of metabolomics pathways (RaMP) pathway clustering algorithm [12].

**Figure 2 cancers-12-02157-f002:**
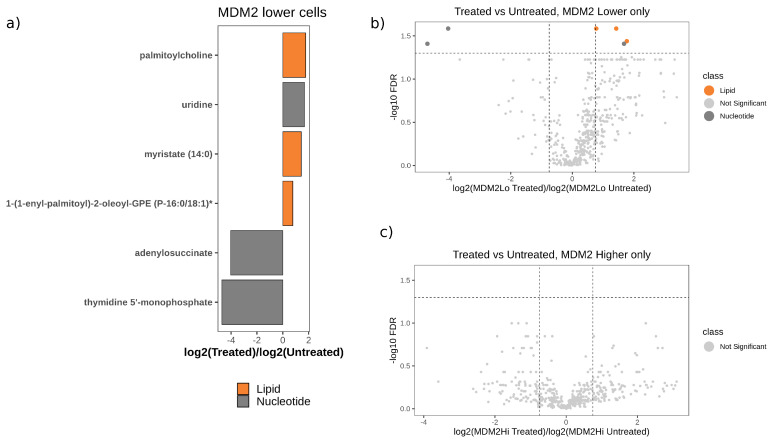
p53 reactivation using an MDM2 binding inhibitor causes lipid upregulation in MDM2 lower cell lines. (**a**) Metabolites altered in DDLPS MDM2 lower amplification cells that are untreated or treated with MDM2 binding inhibition (RG7112) (FDR-adjusted *p*-val < 0.05 and |log2 fold change| > 0.75). (**b**) Volcano plot demonstrating shifts in metabolite levels caused by MDM2 inhibitor treatment in MDM2 lower cells. (**c**) No statistically significant metabolites were identified in MDM2 higher cells.

**Figure 3 cancers-12-02157-f003:**
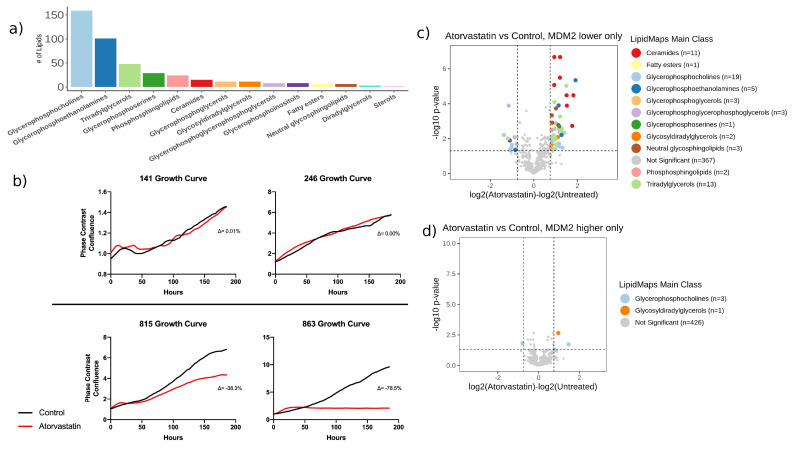
Effects of statin treatment on DDLPS cell lines depends on MDM2 status. (**a**) Coverage of lipids, categorized by their LipidMaps superclass. (**b**) Live cell imaging of DDLPS cell lines demonstrates differential growth patterns stratified by MDM2 status after atorvastatin treatment (average triplicate experiments). (**c**,**d**) We observed a more prominent dysregulation of lipids in response to atorvastatin treatment in MDM2 lower cells (**c**), compared to MDM2 higher cells (**d**).

**Figure 4 cancers-12-02157-f004:**
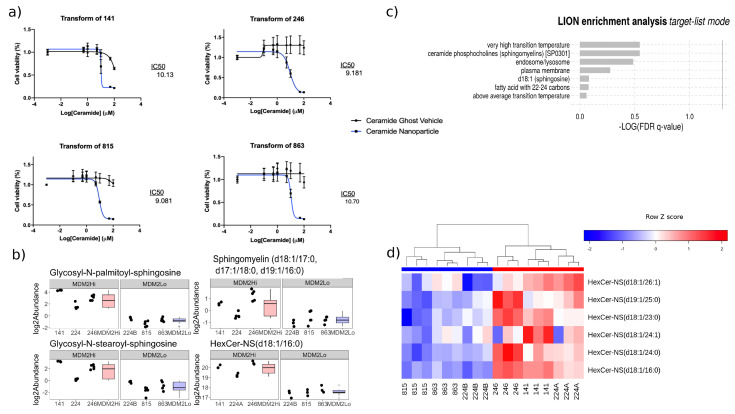
Ceramides are an important differentiator of DDLPS subtypes. (**a**) Treatment with ceramides is lethal to DDLPS cells, regardless of MDM2 status. (**b**) Glycosylated ceramides and lipids in the sphingolipid pathways were amongst the most altered lipids between MDM2 high and low cells in the metabolomic panel. (**c**) LIONenrichment analysis of terms associated with lipids found different between MDM2 higher and lower cells, untreated. (**d**) Hierarchical clustering of cell lines by abundance of all glycosylated ceramides in the lipidomic panel. For column colors, red is MDM2 higher and blue is MDM2 lower.

**Table 1 cancers-12-02157-t001:** Clinical and molecular characteristics of DDLPS cell lines.

Cell Line	MDM2 mRNA Level	MDM2 Amplification Level	Gender	Age
LPS141	473.4	High	M	80
Lipo-246	583.1	High	M	60
Lipo-224A	345.3	High	F	81
Lipo-224B	169.9	Low	F	81
Lipo-815	106.1	Low	M	66
Lipo-863	79.4	Low	M	74

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
