# Peer review of "MDM2-Dependent Rewiring of Metabolomic and Lipidomic Profiles in Dedifferentiated Liposarcoma Models"

_cancers, 2020, doi:10.3390/cancers12082157_

Round 1

Reviewer 1 Report

The manuscript entitled, “MDM2-Dependent Rewiring of Metabolomic and Lipidomic Profiles in Dedifferentiated Liposarcoma (DDLPS) Models” is interesting. Given that DDLPS patients having higher tumoral MDM2 amplification exhibit lower sensitivity to chemotherapeutic agents and have worse outcomes over patients with lower tumoral MDM2 amplification, the authors tested the hypothesis that the changes in DDLPS metabolism could contribute to these observed effects. Overall, the findings indicate that the changes in DDLPS lipid metabolism could possibly be associated with MDM2-dependent differential molecular mechanisms contributing to chemoresistance and poor responses in DDLPS patients. Most of the studies are well-designed, nicely-executed, and presented, and the data are straight forward. There are a few minor comments to address.

  1. What was the rationale of deciding the 72 hours’ time points for the treatment, and not the earlier time points (24 or 48 hours)?
  2. Please reword the sentence wherever application that “…..drug treated with atorvastatin or MDM2 inhibitor for 72 hours”, as …... “drug treated with atorvastatin or MDM2 inhibitor and cultured for 72 hours”.

Author Response

The manuscript entitled, “MDM2-Dependent Rewiring of Metabolomic and Lipidomic Profiles in Dedifferentiated Liposarcoma (DDLPS) Models” is interesting. Given that DDLPS patients having higher tumoral MDM2 amplification exhibit lower sensitivity to chemotherapeutic agents and have worse outcomes over patients with lower tumoral MDM2 amplification, the authors tested the hypothesis that the changes in DDLPS metabolism could contribute to these observed effects. Overall, the findings indicate that the changes in DDLPS lipid metabolism could possibly be associated with MDM2-dependent differential molecular mechanisms contributing to chemoresistance and poor responses in DDLPS patients. Most of the studies are well-designed, nicely-executed, and presented, and the data are straight forward. There are a few minor comments to address.

We thank the reviewer for this feedback. We anticipate that the results of the manuscript will serve as a basis for improved treatment for DDLPS patients with drug-resistant tumors.

What was the rationale of deciding the 72 hours’ time points for the treatment, and not the earlier time points (24 or 48 hours)?

We thank the reviewer for this feedback. The treatments are not cytotoxic agents (directly toxic) and thus peak changes are seen later as metabolic and genomic shifts take more time to manifest. We have updated the manuscript on lines 237-238 to reflect this.

Please reword the sentence wherever application that “…..drug treated with atorvastatin or MDM2 inhibitor for 72 hours”, as …... “drug treated with atorvastatin or MDM2 inhibitor and cultured for 72 hours”.

We thank the reviewer for this feedback. We have edited the sentence on lines 236-237 in the updated manuscript.

Reviewer 2 Report

I think you have an interesting study and I commend you for your work. It will be of interest to the general readership of the journal. 

I just want you to consider and compare your results to what is known in literature about Metabolomic and Lipidomic Profiles in Liposarcoma and how your study will implicate current clinical practice? what do you suggest future research should focus on? 

Author Response

I think you have an interesting study and I commend you for your work. It will be of interest to the general readership of the journal. 

 We thank the reviewer for this feedback. We strove to ensure that our results would be presented in a manner that would appeal to the MDPI Cancers readership.

I just want you to consider and compare your results to what is known in literature about Metabolomic and Lipidomic Profiles in Liposarcoma and how your study will implicate current clinical practice?

We thank the reviewer for this feedback. In the updated manuscript, we have compared our results with previously obtained results studying the metabolome of liposarcoma/DDLPS:

  • On lines 161-165 we have added two references to studies contrasting liposarcoma tumors with healthy tissue, in which the authors identified upregulation of amino acid synthesis and nucleotide synthesis/salvage. We contrast this finding with our observation that MDM2 higher tumor cells also exhibited shifts in these pathways when compared to MDM2 lower, indicating that these processes are perhaps even further dysregulated in the more chemoresistant tumors
  • On lines 198-200 we noted that DDLPS cells are known to have an irregular fatty acyl chain profile compared to healthy cells, and that in our study we noted elevation of several saturated fatty acids in MDM2 higher tumors vs. lower.

We anticipate our study will impact clinical practice in that metabolically driven anti-cancer therapies may potentially spare untargeted toxicities to the patient in this difficult to treat tumor. We are currently exploring the addition of these prioritized metabolic agents to evaluate their utility. We have updated the conclusions section on lines 346-347 to reflect this.

what do you suggest future research should focus on? 

We thank the reviewer for this feedback. In the updated manuscript we have added a statement in the conclusions section on lines 347-349, stating that the future focus of the work will be to validate our findings in human tumor tissue.